# Consumer Impressions of the Safety and Effectiveness of OTC Medicines

**DOI:** 10.3390/pharmacy11020051

**Published:** 2023-03-10

**Authors:** Jeffrey Gordon Taylor, Stephen Ayosanmi, Sujit S. Sansgiry

**Affiliations:** 1College of Pharmacy and Nutrition, University of Saskatchewan, Saskatoon, SK S7N 5E5, Canada; 2College of Pharmacy, University of Houston, Houston, TX 77204, USA

**Keywords:** OTC medicines, safety, effectiveness, product familiarity, public opinion

## Abstract

The public generally believes OTC medicines to be helpful for treating minor ailments. From a survey point of view, that position often originates from feedback obtained when these medicines are considered as one broad category. The objective of the study was to assess the properties of 15 categories of agents across three dimensions—effectiveness, safety, and familiarity. Data were gathered via an online non-random survey in one Canadian province, where residents were asked to consider 15 OTC medicine categories in terms of those dimensions. Five hundred and seventy-five completed surveys were obtained out of 3000 sent. On the 10-point effectiveness scale, values ranged from 5.1 (Athlete’s foot cream) to 7.3 (headache medicine). For safety, the medicines were closely grouped (6.0 to 7.4). Cough syrups for children were perceived as less safe than those for adults. There was a trend in that, as product familiarity grew, so did impressions of safety and effectiveness. The results support other reports where OTC medicines are described as safe and effective, although safety ratings were not particularly high. Responders considered these medicines to generally be higher in safety than effectiveness.

## 1. Introduction

When people experience a minor ailment, the propensity to use an OTC will depend on a host of factors, including what is germane for their situation from sources such as advertisements, friends, and family, or asking a pharmacist. Thereafter, successful use will likely lead to more use, while having less luck (marginal benefit or side effects) will lead to other alternatives.

Generally, consumers perceive OTC medicines to be effective and safe for treating minor ailments [1,2,3,4,5,6,7,8]. An Australian report found that the most important factors to consumers when purchasing OTCs were effectiveness and safety, where personal experience largely predicates those outcomes [9]. Their use has been described as low risk [8,10]. In fact, U.K. data suggest that consumers generally do not consider the risks of OTC use, preferring instead to focus on the benefits [11,12]. Attention was very much on getting better rather than expending effort to evaluate different medicines. Use has even been described as a rather mundane activity [13], where the public assumes that regulatory authorities have the necessary safeguards in place, as was a view held by Canadians [14].

In Canada, when 1000 adult respondents were asked to rate the safety of OTC medicines, there was broad support for them, but stopped short of stating they were *always* safe [6]. About half thought they were also effective either *often* or *always*, while seven percent indicated they were *seldom* or *never* effective.

Much of the research in this area has considered OTC medicines as a broad category. The report just cited, however, takes the analysis a step further and examined safety and effectiveness relative to specific categories of agents [6]. On a scale of *never, seldom, sometimes, often,* and *always*, agents found to be *often/always* safe were as follows: vitamins/minerals (64 percent), cough/cold (60 percent), pain relievers (56 percent), herbals (52 percent), upset stomach/nausea (50 percent), allergy (44 percent), and laxatives (40 percent). The majority of respondents regarded them to be at least *sometimes* safe to use, while in the case of laxatives, one in five considered them to be *seldom* or *never* safe.

For the second parameter, the following agents were considered to be *often* or *always* effective: pain relievers (58 percent), vitamins/minerals (50 percent), upset stomach/nausea (45 percent), cough/cold (42 percent), laxatives (40 percent), allergy (37 percent), and herbals (33 percent). In a similar vein, Americans reported being either *very satisfied* or *somewhat satisfied* with OTC medicines for upset stomach/nausea (94 percent), constipation/diarrhea (94 percent), heartburn/indigestion (93 percent), headache (92 percent), muscle/joint/back pain (89 percent), allergy/sinus (88 percent), cough/cold/flu/sore throat (87 percent), and skin problems (82 percent) [8]. This implies a reasonable level of efficacy, with minimal drawbacks.

The current study expands on assessing specific categories of agents for perceived efficacy and safety. This was part of a larger study covering a broad array of OTC-medicine-related issues.

## 2. Methods

The study was cross-sectional and descriptive in design. Data were gathered via an online survey of residents in one Canadian province. A total of 384 responses were needed for a margin of error of ±5 percent [15]. The source for subjects was a citizen panel database (University of Saskatchewan survey service) of Saskatchewan residents over the age of 18 who had previously volunteered to partake in various surveys. Citizen panels have been used to gain access to various subjects [16], including when involving OTC medicines [17,18]. A randomized mail survey was conducted in this province in the past [19], but difficulties in access to phonebook addresses and rising costs precluded such an approach at this juncture. While subjects were chosen at random from the panel, as a list of volunteers, it was less likely to be reflective of the population as a whole.

Subjects were asked to consider a list of 15 OTC categories across three dimensions. Product inclusion onto that list was based on an iterative process to reflect common agents.

### 2.1. Effectiveness

Effectiveness had been previously defined as the ability of an OTC medicine to produce symptom relief [3]. Questionnaire wording was adapted (and expanded) from various reports [20,21,22], with one using a six-point scale (*not at all effective* to *very effective*) [5]. A 10-point scale has been used to evaluate a decision-support system to improve the safe use of OTC medicines [23]. For the current study, perceived effectiveness was determined by a 10-point scale with worded-anchoring at the poles (1 = *not effective* and 10 = *very effective*) and with the following wording to guide the responder:▪This section looks at the benefit of using a medicine. Medicine effectiveness can include aspects such as it helped with the problem and it worked reasonably fast.▪Feedback on the ones you have used will be relatively easy. Either they helped, or they did not.▪IF you have NOT used some on the list, we would still like your opinion. It may seem unfair to ask for feedback on something you have not used, but ‘hunches,’ ‘best guesses,’ even ‘gut instinct’ is still of interest to us.▪We would like to know how effective are the following OTC medicines?▪On a scale of 1 to 10, a higher number generally means MORE effective. A low number means you do not think the medicine is (or would be) that effective.

### 2.2. Safety

Safety and risk have been assessed in previous reports [1,4,5,24,25]. Reisenwitz evaluated perceived purchase risk on a five-point scale (*very risky* to *not at all risky*) [26], Fielding used a seven-point scale [10], while Lynch had a six-point scale (*no risk at all* to *very high risk*) [5]. Opting to use *safety* rather than *risk*, perceptions were measured on a 10-point scale (poles worded as 1 = *not safe* and 10 = *very safe*) with the following guidance for respondents:▪Safety deals with the downside of taking medicine, such as—side effects, interactions with other medicines, and concern if used by young kids or the elderly.▪Even if you have not used some on the list, we would still like your opinion. Hunches and intuition are still of interest to us.▪We would like to know how safe are the following OTC medicines?▪On a scale of 1 to 10, a higher number generally means GREATER safety—minimal side effects, less worry about interacting with other agents, safe for certain age groups. A low number means you think the medicine is (or would) NOT be that safe.

### 2.3. Product Familiarity

The third scale was created to quantify product familiarity with each agent. Other researchers have examined responder familiarity with OTC medicines relative to risk perceptions and to prescription medicines [27]. The scale used here was a 10-point scale (poles worded as 1 = *not familiar* and 10 = *very familiar*) with the wording presented to responders as follows:▪This section looks at your general familiarity with each type of medicine. By that, we mean—your overall experience with other products like it.▪Our interest goes beyond simply knowing about one product. Considering a medicine for nausea, for example, the focus is on nausea medicines in general, not just the one you might use. How do the various medicines for nausea differ—*effectiveness, side effects, taste*?▪So while a person may know some things about what they use, they may not be too familiar with how other nausea medicines differ across those traits.▪Conversely, IF you just bought such medicine and looked at the different types (*comparison shopped*) before deciding, then you would be more familiar with them than someone who did not do all that.▪With that explanation, we would like to know how familiar are you with each type of medicine listed here?▪On a scale of 1 to 10, a higher number means you are familiar with that category of medicine. Low numbers mean you aren’t that familiar. You may not have used any, nor have you done much comparison shopping on what is available.

Familiarity with an agent was not a prerequisite for providing an opinion on product effectiveness and safety. Even if responders had not used some on the list (as stated above), their opinion was still of interest. Assessing familiarity was still deemed important, however, to estimate how informed they might be with the products when forwarding those opinions.

When opting for a 10-point scale, a mid-point anchored with wording was considered for each scale. This could have manifested, for example, as *neither effective nor ineffective* or *effective half the time*. Interpretations of those could have been *50 percent effective during each use* (symptoms reduced by half) or, conversely, *100 percent effective* for one use but *ineffective* during the next, thereby averaging out to “*effective half the time*”. Given that potential drawback, a mid-point (anchored with wording) was not added, but might still be important as a ‘tipping point’ along a scale during analysis.

For pilot testing, 100 people were invited, with 14 responding. Another 20 citizens provided qualitative feedback (word/phrase difficulties, time to complete, and so on).

Test–retest reliability of the safety and effectiveness scales was measured by manually quantifying the degree of change on the Likert scales from time 1 to time 2. For this, 20 responders were asked to complete the same questionnaire twice (with the second completed a month later). Table 1 and Table 2 show the degree of change in responses over the two time periods.

### 2.4. Survey Distribution

Assuming a response rate of 20 percent, the survey was sent to 3000 provincial residents on the citizens panel in December 2021. Potential subjects were contacted randomly by emails on file, explaining the survey and their rights as participants during first contact, then reiterating the nature of the survey and the questionnaire link (using Voxco survey software) at second contact. Data collection ended four weeks after commencement. The university’s survey service de-identified the data and transferred them to the research team. The study was approved (#2812) by the review board of the University of Saskatchewan in October 2021.

Data were analyzed with descriptive statistics (mean, standard deviation, percentages) and correlations using SPSS software v28. For Pearson r, aggregate scores for familiarity were generated and then correlated to the aggregate scores for safety and efficacy. Aggregate scores were a composite of scoring on the 10-point scale for all 15 categories.

## 3. Results

A total of 575 responses were obtained for a response rate of 19.2 percent. The average age was 63.0 years and the majority (61.6 percent) were female (Table 3). Most (54.8 percent) had a university education, 85.8 percent had no children at home, 41.3 percent considered themselves in very good health, and 53.7 percent lived in larger cities.

On a 10-point scale from *not effective* to *very effective*, the medicines ranged from 5.1 (Athlete’s foot cream) to 7.3 (headache medicine) (Table 4). All 15 types were quite narrowly grouped. Six landed between 5.0 and 5.9 and eight were within 6.0 and 6.9. A value of 5.5 would be considered the mid-point of the scale (although no verbal anchoring was used to reflect that); 11 were above that point. The three most effective agents were headache medicines (7.3), antihistamines (6.9), and fever medicine for a child (6.5). Cough syrups, whether for a child or an adult, garnered similar values.

Regarding safety, the medicines were even more closely grouped, from 6.0 to 7.4 (Table 5). Ten were between 6.0 and 6.9. Cough syrups for children were perceived to be (on average) 0.5 units less safe than such agents for adults and attained the lowest rating on this measure. All medicines were above the mid-point score of 5.5.

Familiarity scores were on the low side, except for headache medicines (6.4) (Table 6). All the rest fell below the mid-point of 5.5, with eight scoring in the three’s or lower.

Pearson r correlations were carried out for familiarity versus safety (r = 0.2, *p* < 0.05), familiarity versus effectiveness (r = 0.4, *p* < 0.05), and effectiveness versus safety (r = 0.6; *p* < 0.05), with each being statistically significant.

## 4. Discussion

Efficacy and safety are obviously critical properties for a medicine to possess. The perception of those parameters for the 15 OTC medicine categories here suggests some support for their use. At the very least, all scored above the scale mid-points. Responders did consider them to be more *safe* than *effective*, however, relatively speaking.

While of interest to see where the agents landed on the effectiveness scale, it is not possible to determine at which point on that scale that an agent attains *effectiveness.* For example, cough syrups for children ranked 0.3 points lower than such syrups for adults. Other than an inter-product comparison now being possible, the 5.3 value for pediatric syrups may still be a respectable outcome. Similarly, while a full 2.0 points lower than headache medicine, it cannot be assumed that a level of *ineffectiveness* is being suggested for the cough syrup. Therapeutic-wise, it is known that cough syrups are not particularly effective, nor are cold sore ointments, but azoles for athlete’s foot are considered effective.

Safety ratings ranged from 6.0 (pediatric cough syrups) to 7.4 (multivitamin). The latter value adds some support to scale validity, in that, of the 15 categories listed, a multivitamin would clearly be the safest. Taken a step further, however, if the 7.4 value sets the high mark for this measure, it is noteworthy that a simple multivitamin would not have garnered a score much closer to 10 at the pole.

Five entities scored values of 7.0 or more on safety. As seen for effectiveness, it would only be conjecture as to what minimum score might be needed for any agent to be deemed *safe* by the public, recognizing of course that the construct will be a gradient. Either way, given the clustering observed for scores, the public seem to consider them all to be relatively similar on this property.

The present study is innovative in that it is one of the few to consider the broad category of ‘OTC medicines’ as smaller groups. It goes a step further than those reports in a few areas. For example, ‘cough/cold medicines’ of other reports was divided into head cold and cough syrups (both pediatric and adult). ‘Skin problems’ were made more specific, such as athlete’s foot and diaper rash. In the American report [8], constipation and diarrhea were considered together, while they were separated in the current report. That said, ‘laxatives’ was not divided into PEG 3350 and senna, nor was ‘headache medicines’ divided into acetaminophen and ibuprofen; there was a balance needed in order to not over-burden responders. Fifteen categories were utilized, with other reports going with seven or eight [6,8].

The current research tends to support other reports where OTC medicines have been described as effective and safe [1,2,3,4,5,6,7,8,14]. It is also important to note, however, that safety was not rated particularly high. This perhaps reflects what was seen years ago (circa 1990) in a national survey, where OTC medicines were considered safe, but not totally so [6]. This may be indicative of a healthy attitude on the part of responders. Likewise, others have found that consumers know there are risks [28] and that these agents should be used with care [29,30].

Somewhat worrisome, OTC products have also been seen by Canadians as not particularly effective (although there was brand loyalty to those they do see as helpful). These agents have been considered as weaker, watered-down versions of prescription drugs, but still generally safe [31]. Even more worrisome is that, in a report of 553 Americans on the acceptability of risk, 75 respondents believed that most OTCs do not have any side effects [32]. Reisenwitz quantified the perceived risk of OTC purchases on a five-point scale (*very risky* to *not at all risky*), where a mean of 3.7 was established, with 35.8 percent stating there was no risk at all to them [26]. This does not bode well for appropriate medicine use.

There was a strong positive correlation between consumer ratings of effectiveness and safety. This finding suggests that consumers who perceive OTC medicines to be effective also think of them as safe. Of course, with drug therapy, this is not always the case. An agent can be very effective but have many side effects and drug interactions. Another agent can be very safe, yet not impart much therapeutic effect. Smaller correlations were seen for product familiarity relative to both safety and effectiveness, with the trend suggesting that, as familiarity rose, so did favorable opinions of them.

The perceptions uncovered during the current study were forwarded with product experience that might be described as limited, other than for headache medicines. Given how common colds are, it was a surprise that head cold medicines and cough syrups both scored below the mid-points on familiarity, as it was to see low familiarity for laxatives, given the average age of the sample. As the sample was older in age, with children no longer home, it was not surprising that familiarity with products intended for pediatric use (fever, coughs, and diaper rash) was low.

One goal for conducting the current study was the potential value in separating the category of OTC medicines into smaller units, thus reducing the drawbacks of a class effect. One report sheds light on this. For a national Health Canada survey, Canadians were asked their opinion of OTC medicine effectiveness (as a category) [6]. Forty-nine percent indicated they are effective *often* or *always.* Later, in this same survey, specific agents considered to be *often* or *always* effective were as follows: pain relievers (58 percent), vitamins/minerals (50 percent), upset stomach/nausea (45 percent), cough/cold (42 percent), laxatives (40 percent), allergy (37 percent), and herbals (33 percent). Thus, in this one case, it appears the class effect led to overall effectiveness being viewed in a more promising light.

The work of Lynch allows comparisons to two other medicinal categories [5]. For two items—(a) *What do you think is the overall level of risk associated with the medicines* (1 = no risk at all) to 6 (very high risk) and (b) *How effective do you think the medicines would be for your condition* (1 = not at all effective to 6 = very effective)—the results were as follows. Perceived efficacy for three types of medicines was as follows: prescribed medicines (4.5), OTCs (3.8), and herbals (3.0). Perceived risk was determined to be as follows: prescribed medicines (3.1), OTCs (2.9), and herbals (2.0). People believed herbal remedies to be less effective, but less risky than OTC and prescribed medicines.

### 4.1. Future Research

Clinical researchers will continue their work on assessing the value of specific agents in each category, such as which second-generation antihistamine or topical intranasal steroid is most beneficial to allergy patients, or which OTC analgesic to use for headache management. Regarding the concepts raised here, how various OTC medicines are used by the public, relative to impressions held, is still in need of examination.

### 4.2. Practical Implications

This work adds to our understanding of patient impressions of OTC medicine safety and efficacy. For the most part, it is good news for such agents, although ratings were not overly impressive. Physicians, nurse practitioners, and pharmacists will continue to be wise to understand how OTC medicines are viewed by potential users.

### 4.3. Study Limitations

The limitations of the current study are ones inherent to any survey research—the distilling of complex human behavior into numerical constructs, specifically what safety and effectiveness means to any one person. Sampling error could have led to responses not reflective of the population under study, as participants were obtained from a sampling frame of volunteers used by the university. The sample reflected the opinions of more educated volunteers (as seen in other survey panel work [18]) who were older and no longer had kids at home. There is concern for the test–retest reliability of the scale measures. For agents such as laxatives, athlete’s foot and diaper rash creams, oral agents for low back pain, heartburn medicines, and cough syrups, two to four units of change were evident from time 1 to time 2.

## 5. Conclusions

The results tend to support other reports where OTC medicines are described as safe and effective. It is important to note, however, that safety ratings were not particularly high, although all scored above the scale mid-point. This may be indicative of a healthy attitude on the part of responders, where consumers know that these agents should be used with care. There was a small trend in that, as product familiarity grew, so did impressions of safety and effectiveness. Physicians, nurse practitioners, and pharmacists must continue to understand how OTC medicines are viewed by potential users.

## Figures and Tables

**Table 1 pharmacy-11-00051-t001:** Degree of change in first and second (different superscripts there) responses for OTC medicine effectiveness.

Agent	Total Subjects with *No Change*	Total Subjects with *1 Unit of Change*	Total Subjects with *2 Units of Change*	Total Subjects with *3–4 Units of Change*	Total Subjects with *5–6 Units of Change*	Total Subjects with *7–9 Units of Change*
Head cold medicine	11 (55%)	2 (10%)	1 (5%)	5 (25%)	1 (5%)	0
Laxative	4 (20%)	2 (10%)	5 (25%)	8 (40%)	1 (5%)	0
Multivitamin	8 (40%)	5 (25%)	2 (10%)	3 (15%)	2 (10%)	0
Antihistamine for allergies	8 (40%)	2 (10%)	3 (15%)	5 (25%)	2 (10%)	0
Athlete’s foot cream	6 (30%)	3 (15%)	4 (20%)	6 (30%)	1 (5%)	0
Diaper rash cream	5 (25%)	6 (30%)	3 (15%)	5 (25%)	1 (5%)	0
Fever medicine for a child	7 (35%)	7 (35%)	2 (10%)	4 (20%)	0	0
Diarrhea medicine	9 (45%)	1 (5%)	4 (20%)	5 (25%)	1 (5%)	0
Low back pain tablet	3 (15%)	6 (30%)	2 (10%)	9 (45%)	0	0
Cough syrup for a child	6 (30%)	5 (25%)	4 (20%)	4 (20%)	1 (5%)	0
Cold sore ointment	9 (45%)	5 (25%)	4 (20%)	2 (10%)	0	0
Drops for an eye infection	7 (35%)	5 (25%)	2 (10%)	6 (30%)	0	0
Headache medicine	7 (35%)	5 (25%)	4 (20%)	3 (15%)	1 (5%)	0
Cough syrup	7 (35%)	5 (25%)	2 (10%)	6 (30%)	0	0
Heartburn medicine	7 (35%)	4 (20%)	2 (10%)	7 (35%)	0	0

**Table 2 pharmacy-11-00051-t002:** Degree of change in first and second responses for OTC medicine safety.

Agent	Total Subjects with *No Change*	Total Subjects with *1 Unit of Change*	Total Subjects with *2 Units of Change*	Total Subjects with *3–4 Units of Change*	Total Subjects with *5–6 Units of Change*	Total Subjects with *7–9 Units of Change*
Head cold medicine	11 (55%)	3 (15%)	3 (15%)	3 (15%)	0	0
Laxative	7 (35%)	5 (25%)	4 (20%)	4 (20%)	0	0
Multivitamin	11 (55%)	4 (20%)	1 (5%)	4 (20%)	0	0
Antihistamine for allergies	9 (45%)	4 (20%)	4 (20%)	3 (15%)	0	0
Athlete’s foot cream	7 (35%)	4 (20%)	3 (15%)	5 (25%)	1 (5%)	0
Diaper rash cream	10 (50%)	2 (10%)	5 (25%)	2 (10%)	1 (5%)	0
Fever medicine for a child	10 (50%)	2 (10%)	5 (25%)	2 (10%)	1 (5%)	0
Diarrhea medicine	8 (40%)	5 (25%)	3 (15%)	4 (20%)	0	0
Low back pain tablet	11 (55%)	3 (15%)	2 (10%)	3 (15%)	1 (5%)	0
Cough syrup for a child	6 (30%)	2 (10%)	4 (20%)	7 (35%)	1 (5%)	0
Cold sore ointment	9 (45%)	3 (15%)	4 (20%)	4 (20%)	0	0
Drops for an eye infection	7 (35%)	5 (25%)	5 (25%)	3 (15%)	0	0
Headache medicine	8 (40%)	7 (35%)	3 (15%)	1 (5%)	1 (5%)	0
Cough syrup	10 (50%)	3 (15%)	2 (10%)	4 (20%)	1 (5%)	0
Heartburn medicine	9 (45%)	3 (15%)	3 (15%)	5 (25%)	0	0

**Table 3 pharmacy-11-00051-t003:** Characteristics of participants.

Characteristics	N	Category	Frequency	Percent
Age (years)	554	Under 20	1	0.2
		20–29	3	0.5
		30–39	22	3.9
		40–49	55	9.9
		50–59	93	16.8
		60–69	218	39.4
		70–79	130	23.5
		80–89	32	5.8
Gender	563	Female	347	61.6
		Male	213	37.8
		Other	3	0.6
Education	564	Some high school	10	1.8
		High school graduate	57	10.1
		Trade/technical school	90	16.0
		Some college/university	98	17.3
		College or university graduate	309	54.8
Number of household children up to 17 years	565	None	485	85.8
		One	29	5.1
		Two	34	6.0
		Three	12	2.1
		Four	4	0.7
		More than four	1	0.3
Health status	564	Excellent	58	10.3
		Very good	233	41.3
		Good	197	34.9
		Fair	62	11.0
		Poor	14	2.5
Place of residence	556	Large city	299	53.7
		Medium city	34	6.2
		Small town	223	40.1

**Table 4 pharmacy-11-00051-t004:** Effectiveness of OTC medicines.

Agent	Impression of Effectiveness	
	N	Mean (Sd)
Head cold medicine	570	6.2 (2.2)
Laxative	570	6.3 (2.4)
Multivitamin	570	5.4 (2.5)
Antihistamine for allergies	568	6.9 (2.1)
Athlete’s foot cream	566	5.1 (2.4)
Diaper rash cream for an infant	569	6.1 (2.6)
Fever medicine for a child	568	6.5 (2.6)
Diarrhea medicine	569	6.3 (2.4)
Low back pain tablet	571	5.8 (2.4)
Cough syrup for a child	567	5.3 (2.4)
Cold sore ointment	567	5.3 (2.4)
Drops for an eye infection	571	6.4 (2.4)
Headache medicine	569	7.3 (2.1)
Cough syrup	569	5.6 (2.2)
Heartburn medicine	568	6.3 (2.3)

**Table 5 pharmacy-11-00051-t005:** Safety of OTC medicines.

Agent	Impression of Safety	
	N	Mean (Sd)
Head cold medicine	568	6.6 (2.1)
Laxative	571	6.2 (2.2)
Multivitamin	569	7.4 (2.3)
Antihistamine for allergies	571	6.8 (2.0)
Athlete’s foot cream	570	7.2 (2.4)
Diaper rash cream for an infant	571	7.3 (2.3)
Fever medicine for a child	570	6.4 (2.2)
Diarrhea medicine	570	6.6 (2.1)
Low back pain tablet	568	6.3 (2.1)
Cough syrup for a child	569	6.0 (2.3)
Cold sore ointment	570	7.2 (2.2)
Drops for an eye infection	570	6.8 (2.2)
Headache medicine	569	7.0 (2.0)
Cough syrup	571	6.5 (2.1)
Heartburn medicine	569	6.8 (2.1)

**Table 6 pharmacy-11-00051-t006:** Product familiarity.

Agent	Product Familiarity	
	N	Mean (Sd)
Head cold medicine	571	5.3 (2.6)
Laxative	568	3.8 (2.7)
Multivitamin	568	4.9 (2.7)
Antihistamine for allergies	569	5.1 (2.8)
Athlete’s foot cream	568	2.6 (2.2)
Diaper rash cream for an infant	569	3.0 (2.6)
Fever medicine for a child	567	3.5 (2.8)
Diarrhea medicine	571	3.9 (2.6)
Low back pain tablet	568	4.8 (2.9)
Cough syrup for a child	569	3.3 (2.6)
Cold sore ointment	569	3.6 (2.8)
Drops for an eye infection	568	3.9 (2.7)
Headache medicine	570	6.4 (2.6)
Cough syrup	567	4.8 (2.5)
Heartburn medicine	569	4.8 (2.9)

## Data Availability

The data presented in this study are available on request from the corresponding author.

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
