# Peer review of "Consumer Impressions of the Safety and Effectiveness of OTC Medicines"

_pharmacy, 2023, doi:10.3390/pharmacy11020051_

Round 1

Reviewer 1 Report

Title: Consumer Impressions of the Safety and Effectiveness of OTC Medicine. - Please include the word familiarity and information about Where and When?

Abstract

-          15 individual agents; distribution per pharmacotherapeutic groups?

-          via an online 14 survey; timeframe?

-          “Five hundred and seventy-five completed surveys were obtained.”; % of valid questionnaires?

-          “Familiarity scores were on the low side.” Please explain the concept of “familiarity”. Low side?

Key words: please consider some MeSH terms.

Introduction

-         - Please cite more studies in introduction, such as reviews papers related to the present topic. For instance, see https://pubmed.ncbi.nlm.nih.gov/?term=patient+perception+over+the+counter&filter=datesearch.y_5.

 -       - Why is the present study innovative and relevant for an international audience?

-       -    Please briefly explain these three concepts in introduction: effectiveness of medicines, safety of medicines and familiarity with medicines.

-          - “In Canada, when respondents were asked to rate the safety of OTC medicines, there was broad support for them, but stopped short of stating they are always safe.6 About half thought they were also effective either often or always, while seven percent indicated they were seldom or never effective.” In Canada? Or in one study carried out in Canada? Please give more details. Why are some words underlined?

-         .  “The current study expands on assessing specific agents for perceived efficacy and safety.” What about familiarity? Why was familiarity not included in study objectives?

2. Methods

- Some issues are missing:

- Survey: Development? What studies supported their development? Validation? Involved experts? When was the survey administered? Please fundament the choice of a 10-point scale? % of valid questionnaires/replies?

- Please create a subheading about the survey in methods.

- Please present the original of the survey in a supplementary file.

- Sampling strategy?

- online survey: How was the survey administered? (e-mail? Facebook?); self-administration? Administration time? Where questions blocked? Please give more details methods must be reproductible.

- OTC medicines: What OTC medicines? Distribution per pharmacotherapeutic groups? Why these OTC medicines (and not others)?

- Please also create specific subheadings for OTC medicines and statistical methodologies. Please correctly cite SPSS. Please give more details about descriptive statistics (mean; average; proportions?). Why inferential tests were not applied? Please cite studies of similar research using the same statistical methodology.

- Authors have Pearson r correlations in results. Please explain Pearson correlations in methods. Please ensure that all statistical methodologies are explained in methods.

- Ethical issues? Confidentiality of data? Will replies be destroyed? When? Approval number? Informed consent?

Results

-     - Results are too compact. For instance, create the following subheadings in results: participants; efficacy; safety and effectiveness.

-   - Please briefly describe Tables 1 to 6 in methods highlighting the main findings.

-          - Please create some figures and/or flowcharts.

Discussion

-          - Please cite more related studies, such as reviews in discussion.

-          - Please discuss all topics in the same order of methods and results.

-        -  Please ensure an exhaustive discussion of all study findings.

-   - Why is the present study innovative and relevant for an international audience?

-          - What is the contribution of the present study to the state-of-the-art?

-         -  What were the key findings from the present study?

-          Please create the following subheadings at the end of discussion: future research; practical implications and study limitations. Please discuss the possibility of study bias in study limitations.

-          “One premise for conducting the current study was the potential value in separating the category of OTC medicines into actual agents, thus reducing the drawbacks of a class effect.” This type of information should be placed in methods.

-          It seems some citations are missing in discussion.

-          “as participants were obtained from a sampling frame of volunteers used by the university.” This type of information should also be placed in methods. Paid volunteers? Please clarify.

Conclusion

-          Conclusion must reply to study objectives. Other information should be placed in discussion.

Tables and references:

-          Please check the format of Tables and references in instructions for authors and published papers from Pharmacy.

Author Response

As attached.

Reviewer 2 Report

Thank you for the opportunity to review this very interesting manuscript on the perceptions of OTC products.  Overall, the study is well-designed and provides interesting data.  I offer the following suggestions to enhance the manuscript

Please comment on the low response rate.  Could this low response lead to bias?

Please comment on the study population.  Is the sample indicative of the typical consumer of OTC products?  What is the external validity?

Line 213.  Please elaborate, is 0.6 considered a strong correlation?  https://link.springer.com/article/10.1057/jt.2009.5

Author Response

Please see the attachment, with the other reviewer to follow tomorrow.

Round 2

Reviewer 1 Report

-          Dear authors, Tanks for all updates. The quality of the paper has been improved. I have rated the paper with major revisions because the introduction is insufficiently developed. A section about the applied statistical methodologies is also lacking.

-          Please proofread the paper once more. For instance, authors have “15” in upper script in line 70, page 2. Moreover, some of the references in introduction are also in upper script.

- -         - Please cite more studies in introduction, such as reviews papers related to the present topic. For instance, see https://pubmed.ncbi.nlm.nih.gov/?term=patient+perception+over+the+counter&filter=datesearch.y_5.

-          Authors response: “I don’t think we missed any key reports in our work, as we used over 300 citations in the complete report as a dissertation. I don’t think articles can cite all the papers relevant to an area, so we just listed the main ones of relevance. I will be happy to add more refs and they are now in there, but if you want to state which ones specifically, I can add others too. I don’t recall any review papers on the safety and efficacy of OTC medicines tho.”

Reply: I disagree with authors. In general, readers will not consult the thesis. The introduction comprises 8 references. None of these references is a review. In general, an introduction should comprise at least 15 references. Please consult the instructions for authors and check the section about introduction. Introduction is insufficiently developed.

Instructions for authors: “Introduction: The introduction should briefly place the study in a broad context and highlight why it is important. It should define the purpose of the work and its significance, including specific hypotheses being tested. The current state of the research field should be reviewed carefully and key publications cited. Please highlight controversial and diverging hypotheses when necessary. Finally, briefly mention the main aim of the work and highlight the main conclusions. Keep the introduction comprehensible to scientists working outside the topic of the paper.”

-          At least some of the new cited references in introduction should be explained in discussion.

-          Please briefly explain these three concepts in introduction: effectiveness of medicines, safety of medicines and familiarity with medicines.

Authors: “I am not sure that would be the usual approach out there. Readers in healthcare will surely understand what medicine safety and efficacy is, and product familiarity would not be that abstract either. We would opt to keep the definitions in the Methods. If added earlier, we would simply be repeating what is currently in Methods.”

Reply: please note that not all the readers are health professionals. At least, the concept of familiarity must be explained (in introduction or in methods) because this concept is subjective (or may be subjective). What is familiarity with an OTC? Please define/explain.

Methods

-          Ok authors have not validated questionnaires, but was a pre-test carried out (at least)? The pre-test is relevant to understand respondents’ comprehension. Have participants doubts about the questionnaire?

-          “I don’t think any report out there has validated their scale for efficacy nor safety.” In my opinion, this sentence is not enough to explain the inexistence of a questionnaire validation.

Authors reply: “Validation? I don’t think any report out there has validated their scale for efficacy nor safety. We did not validate them either, as they did not really hit the level of scale validation one might need for more complex constructs. But it is fair criticism. That said, we did add the aspects of Degree of Change in Responses (Table I) to let readers know how things looked for reliability at least. We did validation work, reliability measures, non-response bias, early vs late responders, Cronbach’s alpha etc for other aspects of the project, and that has been reported elsewhere (Innov Pharm 2022).”

-          Distribution per pharmacotherapeutic groups? Why these OTC medicines (and not others)? Please give a brief explanation in the text.

-          A section about the applied statistical methodologies is lacking. These type of arguments “But a researcher does what test is most appropriate” or “Likert scales are extremely common” are not enough to justify the inexistence of a section about the applied statistical methodologies.

-          Authors: “I will leave that up to the editor b/c for a section of this length, I am not sure sub-headings are needed to help the reader get through the data.”

-          Reply: authors are advised to consult published papers from Pharmacy, which, in general, use subheadings in the section of results. The use of subheadings will increase text readability.

-          The objective of the study was to assess the properties of 15 categories of agents across two dimensions – effectiveness and safety.

Conclusion: “The results tend to support other reports where OTC medicines are described as safe and effective. It is important to note, however, that safety ratings were not particularly high, although all scored above the scale mid-point. This may be indicative of a healthy attitude on the part of responders, where consumers know that these agents should be used with care. There was a small trend in that as product familiarity grew, so did impressions of safety and effectiveness.”

Authors response: “The conclusion has 4 sentences, with safety and efficacy and familiarity listed out, and the key impact of the data. Not sure what in there would be considered out-of-place and therefore must be reverted to the Discussion?”

Reply: In my opinion, it is not usual to make explicit mentions to study findings in conclusion, such as “The results tend to support” or “that safety ratings were not particularly high, although all scored above the scale mid-point”. In my opinion, study conclusion should be rewritten. Please exclusively reply to study objective in conclusion.

Other comments: I highly recommend that authors carefully read the replies to the reviewer, before elaborating and submitting cover letter.

Author Response

Hi Editors and/or Reviewer 1,

I have attached my feedback on the comments. Hopefully I have attained a level of satisfaction with them. 

Of note, I sent Mia the latest version as a WORD file, but of course would make changes with track-changes at the appropriate time to the copy that already is in your system. 

J Taylor
